# Constant-time Quantum Algorithm for Homology Detection in Closed Curves

Nhat A. Nghiem[1*], Xianfeng David Gu[2,3], Tzu-Chieh Wei[1,4,5]

**1** Department of Physics and Astronomy, State University of New York at Stony Brook, Stony Brook, NY 11794-3800, USA
**2** Department of Computer Science, State University of New York at Stony Brook, Stony Brook, NY 11794, USA
**3** Department of Applied Mathematics & Statistics, State University of New York at Stony Brook, Stony Brook, NY 11794, USA
**4** C. N. Yang Institute for Theoretical Physics, State University of New York at Stony Brook, Stony Brook, NY 11794-3840, USA
**5** Institute for Advanced Computational Science, State University of New York at Stony Brook, Stony Brook, NY 11794-5250, USA
* nhatanh.nghiemvu@stonybrook.edu

February 28, 2023

## Abstract

Given a loop or more generally 1-cycle $r$ of size **L** on a closed two-dimensional manifold or surface, represented by a triangulated mesh, a question in computational topology asks whether or not it is homologous to zero. We frame and tackle this problem in the quantum setting. Given an oracle that one can use to query the inclusion of edges on a closed curve, we design a quantum algorithm for such a homology detection with a constant running time, with respect to the size or the number of edges on the loop $r$, requiring only a single usage of oracle. In contrast, classical algorithm requires $\Omega(L)$ oracle usage, followed by a linear time processing and can be improved to logarithmic by using a parallel algorithm. Our quantum algorithm can be extended to check whether two closed loops belong to the same homology class. Furthermore, it can be applied to a specific problem in the homotopy detection, namely, checking whether two curves are *not* homotopically equivalent on a closed two-dimensional manifold.

# 1  Introduction

Topology and geometry are among the most classic and fundamental areas in pure mathematics. Despite being abstract and sometimes counter-intuitive, their role in both science and engineering has been impactful. In physics, ideas from topology have provided a framework that explains phases of matter beyond Landau's symmetry-breaking theory [1], such as the so-called topological phase of matter, and offered the prospect of fault tolerance with schemes of topological quantum computation [2]. In engineering and applied sciences, for example, topological data analysis [3,4] employs techniques from topology to analyze and identify patterns or shapes of high-dimensional data. The foundation of computational conformal geometry has also been laid out [5–7], providing valuable tools for applications, such as mechanical designs, medical imaging, computer vision, and so on.

At the same time, the notion of quantum computers [8–11] has generated an entirely new frontier in computational science. By harnessing the enigmatic properties of quantum mechanics, such as entanglement and superposition, quantum computers possess the potential to handle specific challenging computational problems that are not thought to be efficient within reach of classical computers. Some famous classic problems including factorization [9], unstructured search [10], and linear system solvers [12], etc. The use of quantum computers has also been extended to the modern context, such as machine learning and data science [13–23]. In Ref. [24], quantum computational techniques were applied to topological data analysis; specifically, a quantum algorithm was constructed to estimate Betti numbers of a given simplicial complex, which yields an exponential speedup compared to classical algorithms.

Inspired by the development in both quantum computation and computational topology and geometry, here, we attempt to apply quantum approaches to advancing tools and solving problems in topology and geometry. As a small step, we consider the problem of detecting the homology class of closed curves or 1-cycles. As explained below, the ability to detect

homologically trivial curves provides a means to a related problem in the homotopy detection. The precise statement of our main problem is as follows: given a closed surface, represented as a triangular mesh, and a closed curve (or loop) on the surface, we want to know whether or not the curve is homologous to zero, i.e., trivial homologically. Remarkably, given a sufficient number of qubits and the oracle that queries the curve, our algorithm can detect a given closed curve's homology class of with certainty at a constant time complexity. We remark that reducing efficient classical solutions to even more efficient quantum algorithms is also of interest from the complexity perspective. One such example is the 2D hidden linear function problem [25].

The structure of this paper is as follows. First in Sec. 2, we introduce and clarify some terms/terminologies that are relevant to our subsequent construction. We mention some assumptions that our algorithm relies on in Sec. 3. In Sec. 4, we explicitly construct the quantum algorithm for detecting the homology class of closed curves. We conclude with some discussions and an outlook in Sec. 5. Along the way, we have also developed an efficient algorithm that creates a uniform superposition of computational basis states that are in a consecutive sequence; see the Appendix.

## 2 Preliminaries

### 2.1 Overview of essential concepts in homology and cohomology

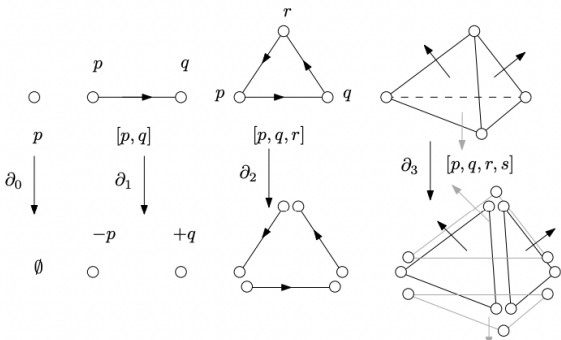

Figure 1: Illustration of chains and the boundary operation. The subscript of each boundary operator $\partial$ indicates which chain space it acts on (from the higher to lower order).

Here, we give a quick overview of homology and cohomology concepts. In the language of algebraic topology, see, e.g., Ref. [26], we can treat a (discrete) surface as a simplicial complex, consisting of different dimensional simplices. The linear combinations of simplices, in which the coefficients belong to some ring $R$, form chains. Two common rings that are usually used are $\mathbb{Z}_2$ and $\mathbb{Z}$. A curve is then a 1-chain in the discrete setting. The set of $k$-chains forms a linear vector space called the chain space, denoted by $C_k$. There is a map between two chain spaces $C_k$ and $C_{k-1}$ called the *boundary map* $\partial_k : C_k \to C_{k-1}$, which maps a $k$-chain to its boundaries, as illustrated in Fig. 1.

A $k$-chain $\sigma_k$ is called *closed* if $\partial_k \sigma_k = 0$, and it is also referred to as a $k$-cycle. A loop is

such an example. A $k$-chain $\sigma_k$ is called *exact* if there exists a $(k+1)$-chain $\sigma_{k+1}$ such that

$$\sigma_k = \partial_{k+1}\sigma_{k+1}.$$

We note that exact chains are closed by virtue of an important property of boundary map [6]: $\delta \circ \delta = 0$. Two closed $k$-chains are equivalent if they differ by an exact $k$-chain. The equivalence relation divides $k$-chains into the so-called *homology classes*, denoted as $H_k(M,\mathbb{Z})$ (where we assume the ring is $\mathbb{Z}$, for simplicity). Homology theory reflects the algebraic structure (note that connectivity between points is the key) of a given simplicial complex via the corresponding homology classes, or more precisely, homology groups.

   On the other hand, cohomology is dual to homology. Given a $k$-chain $\sigma$, a $k$-cochain $w$ maps it to a real number $w(\sigma) \in R$. We can think of cohomology as the association of elements in homology with a real number. Due to the linearity of their vector spaces, it is convenient to use a basis in homology and the dual basis in cohomology groups. For a closed surface of genus $g$, the dimension of homology/cohomology basis is $2g$. Given a homology basis $\langle h_1, h_2, \ldots, h_{2g}\rangle$, the corresponding cohomology basis, denoted by as $\langle \Omega_1, \Omega_2, \ldots, \Omega_{2g}\rangle$, can be constructed, such that

$$\int_{h_i} \Omega_j = \delta_{ij}.$$

   An arbitrary closed curve $\gamma$ is homologous to zero if and only if

$$\int_{\gamma} \Omega_\alpha = 0, \quad \forall \alpha \in \{1, 2, \ldots, 2g\}. \tag{1}$$

More concretely, if the closed curves $\gamma$ has $L$ oriented edges labeled by $\vec{e}_j$, such that

$$\partial_1 \left(\sum_{j=1}^{L} \beta_j \vec{e}_j\right) = \sum_{j=1}^{L} \beta_j(\partial_1 \vec{e}_j) = 0,$$

where $\beta_j \in \mathbb{Z}$. Then the above condition of being homologically zero means that for every $\alpha$, we have:

$$\Omega_\alpha(\gamma) = \sum_{j=1}^{L} \Omega_\alpha(\beta_j \vec{e}_j) = \sum_{j=1}^{L} \beta_j \Omega_\alpha(\vec{e}_j) = 0. \tag{2}$$

This formulation is more suitable for constructing of our quantum algorithm, presented later.

   The run time for a naive classical algorithms is $\mathcal{O}(2g \cdot L)$ [5], where $L$ is the number of edges (1-chains) on the closed curve $r$, and, therefore, the larger the loop is, the more computational time is required [7]. This seems reasonable, as whether a loop is homologically zero is a *global* and topological property. We remark that the key step is to perform the summation along the curve r (see equation 2), so the running time can be improved to logarithmics by using parallel algorithm (for example, we can divide the edges into groups and sum them individually, then summing over). However, this parallel algorithm increases the memory usage by $\mathcal{O}(L)$ (naive approach takes constant memory). Regarding oracle usage, those above two classical approaches require $\Omega(L)$ usages, as classical algorithm is supposed to query all the edges to completely specify the loop. However, we will provide a quantum algorithm below and show that it can determine the homology property of a given closed curve (specified by some oracle $\mathcal{O}_r$ that only checks local properties) with running time $\mathcal{O}(2g)$, where $g$ is the genus of the surface, and a single usage of oracle. Thus, as we will show, if there are sufficient

ancillary qubits in the phase estimation step in our algorithm, then the homology class of the given closed curve could be determined in constant time, with respect to the size L of the curve.

## 2.2 Mesh Data Structure

Despite the fact that surfaces are continuous, in real applications, such as digital geometry processing, they are usually represented as a triangulated polyhedron surface, namely a triangular mesh; see, e.g. Fig. 2 for illustration. A mesh $M = (V, E, F)$ consists of sets of

(a)

(b)

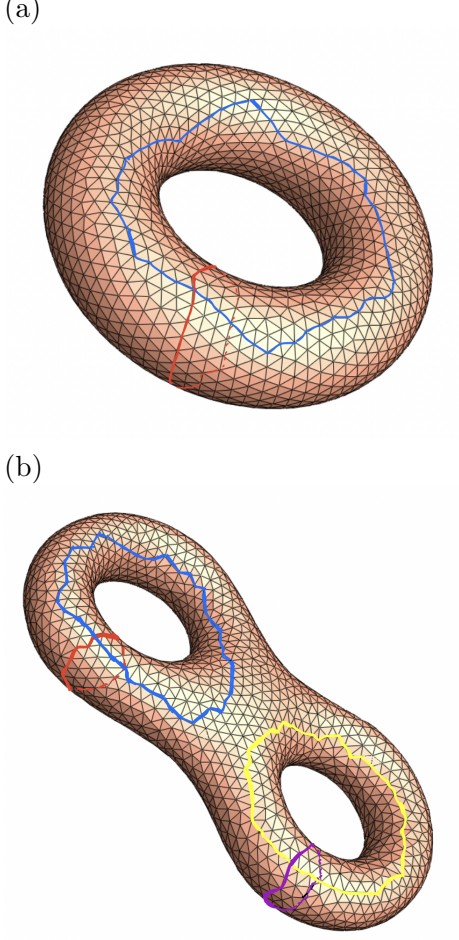

Figure 2: Examples of genus-1 and genus-2 oriented surface, represented as a triangular mesh, with the corresponding homology basis. (a) Top figure: two 1-chains (red and blue) are homology basis. (b) Bottom figure: four 1-chains (red, blue, yellow, purple) are homology basis.

vertices $V$, edges $E$, and faces $F$. We can think of it as a graph with vertices and edges that connect vertices. In terms of the algebraic topology, it is precisely a two-dimensional simplicial complex.

    We denote the set of vertices as $V = \{v_0, v_1, ..., v_M\}$. For a given edge $e_i$ that connects two vertices $v_j, v_k$, for instance, a half-edge is an oriented edge $\vec{e_i} = [v_j, v_k]$, which implies the

orientation $v_j \to v_k$. We simply denote $-\vec{e_i} = [v_k, v_j]$ for the reverse order of the edge and vertices and use the vector notation $\vec{e_i}$ just to emphasize the orientation. The importance of half-edges is apparent for the purpose of computation with cohomology. Suppose $w$ is a 1-cochain, then for a given 1-chain (edge) $e_i$ connecting $v_j$ and $v_k$, we have $w([v_j, v_k]) = -w([v_k, v_j])$, i.e., $w(-\vec{e_i}) = -w(\vec{e_i})$. An exact 1-cochain can be constructed from some 0-cochains $f$ by taking the coboundary map $d$:

$$df([v_j, v_k]) = f(v_k) - f(v_j).$$

Using this, we explain in Appendix A how the cohomology basis (which is exact) can be constructed simply by subtracting some carefully initialized 0-form.

## 3  Setup for the quantum approach

Here we describe several assumptions for the input of our quantum algorithm to be described in the next section.

**Preprocessing Mesh:** We remark that we do not consider the problem of generating triangulated mesh here. We assume that such a step is done classically.

**Homology and Cohomology Basis**: Similar to classical algorithms, we assume that the homology basis has been pre-computed classically, and the cohomology basis has also been constructed accordingly. In other words, we have predefined values for each edge (with orientation) of the mesh. In fact, for most such half-edges, these values are 0, except for a few that connect those neighboring points on and off the homology basis, which can have values 1 or -1, dependent on the orientation of the half-edge; see also Appendix A. For each cohomology basis $\Omega_\alpha$, denote the number of those non-zero values as $c_\alpha$. We further assume that two curves in the homology basis only intersects at no more than one vertex, which means that the two cohomology basis elements have nonzero value at no more than one common edge. Those conditions can always be satisfied, using a result of [27]. Moreover, we assume the number edge $E$ is known and that the value of $c_\alpha$ (number of edges on which $\Omega_\alpha$ has a nonzero value) is known for all $\alpha$ via classical pre-processing.

**Encoding of Cohomology Basis in a Quantum State**: As described above, we can think of cohomology as the association of each half-edge (1-simplex) with a real number. Therefore, in principle, we could map these values to some quantum state with corresponding entries using $\sim \mathcal{O}(\log_2(E))$ qubits, where $E$ is the total number of edges on the mesh. We remark that the values on most edges are zero, except for those edges that connect points on the homology basis to their neighboring points that are not on the basis [7]; see also Appendix A. We first need to map such a cohomology basis to the quantum state. A vector corresponding to a basis $\Omega_\alpha$ (with $\alpha = 1, \ldots, 2g$) has the form

$$\vec{\Omega}_\alpha = [\omega_1^{(\alpha)}, \omega_2^{(\alpha)}, \ldots]^T = [\pm 1, 0, ..., \pm 1, ...0]^T,$$

where the component in the vector represents the value $\Omega_a(\vec{e_j}) = \omega_j^{(\alpha)}$ on each edge. By choosing the orientations of the edges appropriately and using $\Omega_\alpha(-\vec{e_j}) = -\Omega_\alpha(\vec{e_j})$, we can

make the nonzero entries be uniformly +1. We can also re-arrange the edge labeling so that $\vec{\Omega}_1 = [1, 1, \ldots, 1, 0, \ldots, 0]^T$ and $\vec{\Omega}_2 = [0, \ldots, 1, 1 \ldots, 1, 0, \ldots, 0]^T$, etc., where if $\vec{\Omega}_2$ overlaps with $\vec{\Omega}_1$ on an edge, we will arrange such that the particular overlapped edge corresponds to the last nonzero entry in $\vec{\Omega}_1$ and the first nonzero entry in $\vec{\Omega}_2$; otherwise, the first nonzero entry of $\vec{\Omega}_2$ will be the next entry to the last nonzero one in $\vec{\Omega}_1$.

Thus, for simplicity, we shall fix the orientation of edges such that all the nonzero entries are +1. That is those oriented edges $\vec{e}_i$ are regarded as the 'positive' half-edges, $\Omega_\alpha(\vec{e}_i) = +1$. For any edge $e_i$, we can label it by some computational basis state, which we also denote as $|e_i\rangle$ (e.g. $|0101\ldots\rangle$) and its orientation will be flagged by another ancillary qubit,

$$|\vec{e}_i\rangle \equiv |0\rangle |e_i\rangle, \tag{3}$$

$$|-\vec{e}_i\rangle \equiv |1\rangle |e_i\rangle. \tag{4}$$

One can easily see that the quantum state corresponds to each of the vector $\vec{\Omega}_\alpha$ (normalized to 1) is just a uniform superposition of some basis states that encode the edges that are labelled in a consecutive way, and this superposition can be prepared with a very low cost, as we elaborate further in Appendix C. In the simplest scenario, for example, the number of non-zero values is a power of 2, and all those entries lie in "perfect" locations that could give us a simple, convenient way to prepare the corresponding quantum cohomology basi, using Hadamard gates $H^{\otimes n}$ only. For example, we consider the following vector of represent their cohomology basis values on four edges:

$$\vec{x} = [1, 1, 0, 0]^T,$$

whose corresponding quantum state $|x\rangle$ is simply $|0\rangle|+\rangle$, which can be prepared simply by applying $I \otimes H$ to $|0\rangle|1\rangle$.

However, we may not have those nice conditions in a general mesh. One certainly has the freedom to relabel edges. Still, one may need to locally modify the mesh so that the number of edges (with a nonzero value) in each cohomology basis is a power of two. Even if this is the case, the nonzero entries may not necessarily range from (in the binary representation) $|0000\ldots0ab..c\rangle$ to $|0011..1ab..c\rangle$, and their superposition cannot be obtained simply by applying Hadamard gates. But even if it were the case, how could one prepare a corresponding superposition? One could use a two-step procedure of (1) Hadamard gates to create a uniform superposition from $|00\ldots000..0\rangle$ to $|0011..100..0\rangle$ and then (2) a quantum adder [28] to add the appropriate shift $|00\ldots0ab..c\rangle$, which is efficient.

In Appendix C, we show that even without modifying the mesh structure, by choosing the appropriate labeling of edges (as described above), we can efficiently prepare all the cohomology basis states, as stated below.

**Claim 1** *A quantum state associated $|\Omega_\alpha\rangle = \sum_j \omega_j^{(\alpha)} |e_j\rangle / \sqrt{c_\alpha}$ with the cohomology basis with $c_\alpha$ non-zero amplitudes can be prepared using a circuit of depth $\mathcal{O}(\log(c_\alpha))$.*

As we will point out later, given a mesh with $M$ points, then the number of edges used for each cohomological basis is small, i.e., $c_\alpha \ll M$. The cost to prepare the cohomology basis state is negligible and is of $\mathcal{O}(\log(c_\alpha))$, as detailed in Appendix C ,and thus it does not incur substantial computational time.

**Quantum oracle for loop specification:** We remark that in this problem, we are interested in the homology property of a given closed curve $r$. The orientation of the curve is also

important. We have assumed that certain orientation for each edge is fixed, so that all the cohomology bases have a uniform sign in their nonzero components, as discussed earlier. Classically, we can specify the loop $r$ by listing all its half-edges. Naively, on quantum computers, we would like to have a single quantum state in a superposition of basis states that encode the edges on $r$. But this requirement is too strong. Instead we would like to have a quantum oracle that we can use to probe the relation of an edge $e$ to the loop $r$. More precisely, it should be a half-edge $\vec{e}$ (where the arrow indicates a certain orientation, chosen for convenience as positive). Its representation by a quantum state is illustrated in Eq. (3) and the corresponding half-edge with the negative orientation $-\vec{e}$ is represented in Eq. (4). The unitary $\mathcal{O}_r$, or oracle, associated with a given closed curve $r$, is designed so that, when queried with an input of an edge $|\vec{e}_i\rangle$ and an ancillary qubit,

$$
\mathcal{O}_r \, |\vec{e}_i\rangle \, |0\rangle = \begin{cases} |\vec{e}_i\rangle \, |1\rangle & \text{if half-edge } \vec{e}_i \in r \\ |\vec{e}_i\rangle \, |0\rangle & \text{if half-edge } \vec{e}_i \notin r \end{cases} \tag{5}
$$

Essentially, the function of the oracle is to check whether a given half-edge $\vec{e}_i$ is on the curve $r$. However, the above functioning of the oracle did not take into account the possibly multiple appearances of some edges on the loop. If the above oracle is given, then we can only deal with the case where the coefficients of chain space belong to $\mathbb{Z}_2$. If we have access to the oracle that performs the following operation instead:

$$
\mathcal{O}_r \, |\vec{e}_i\rangle \, |00..0\rangle = \begin{cases} |\vec{e}_i\rangle \, |a\rangle & \begin{array}{l} \text{if half-edge } \vec{e}_i \in r \text{ and} \\ \text{appears } a \text{ times,} \end{array} \\ |\vec{e}_i\rangle \, |00..0\rangle & \text{if half-edge } \vec{e}_i \notin r, \end{cases} \tag{6}
$$

where the binary string $a = a_{n_s-1}...a_1 a_0$ denotes the number of times the given half-edge $\vec{e}_i$ appears on $r$. (In fact, in the above, the top line already includes the bottom line as a special case with $a = 0$.) Then we can deal with $\mathbb{Z}_{2^{n_s}}$ coefficients. Note that if $a = 0$ or 1, then we recover the ($\mathbb{Z}_2$) oracle given by Eqn. 5. We do not expect our algorithm can work in the case where an edge appears infinite times (for example, a loop winding around infinitely). Therefore, a reasonable assumption is that each half-edge only appears a bounded number of times less than $2^{n_s} = K$ (with $K$ being a constant) or, alternatively, the oracle only checks the number of times modulo $K$.

## 4    Quantum Algorithm

In this section, we present our main result: a quantum algorithm for the homology detection of a loop. We will employ a Hadamard test procedure and combine it with quantum phase estimation to evaluate whether the sum described in Eq. (2) for each cohomology basis on the curve $r$ is zero or not.

### 4.1    A Hadamard test procedure for estimating phases

Our primary method is built upon the techniques in Refs. [10], [29] and [30], which were recently employed in the setting of quantum neural networks, e.g., see Ref. [31]. We quickly review the routine as it will be generalized to form the main ideas of our quantum algorithm.

Suppose we have some unitary $U$ that generates the following $(n+1)$-qubits state by $|0\rangle^{\otimes n+1}$:

$$U|0\rangle^{\otimes n+1} = |\phi\rangle = \frac{1}{\sqrt{2}}(|+\rangle|x\rangle + |-\rangle|y\rangle), \tag{7}$$

where $|x\rangle, |y\rangle$ are some $n$-qubits states. We can then use it to construct the Grover-like unitary $G = U S_0 U^\dagger (Z \otimes I^{\otimes n})$ [10, 29, 31], where $S_0 = I^{\otimes n+1} - 2(|0\rangle\langle 0|)^{\otimes n+1}$, such that $|\phi\rangle$ can be written as a linear combination below,

$$|\phi\rangle = \frac{-ie^{i\theta}}{\sqrt{2}}|w_+\rangle + \frac{ie^{-i\theta}}{\sqrt{2}}|w_-\rangle, \tag{8}$$

in which $|w_\pm\rangle \sim |0\rangle(|x\rangle + |y\rangle)/(\||x\rangle + |y\rangle\|) \pm i|1\rangle(|x\rangle - |y\rangle)/(\||x\rangle - |y\rangle\|)$ are two eigenstates of $G$,

$$G|w_\pm\rangle = e^{\pm i2\theta}|w_\pm\rangle, \tag{9}$$

and the angle $\theta$ satisfies the relation

$$\sin(\theta) = \frac{\sqrt{1 + \Re\langle x|y\rangle}}{\sqrt{2}}, \tag{10}$$

or equivalently $\cos(2\theta) = -\Re\langle x|y\rangle$. This relation suggests that the real part of the overlap $\langle x|y\rangle$ is encoded in the phase $\theta$, which is related to the two eigenvalues. We also note that $e^{-i2\theta} = e^{i(2\pi - 2\theta)}$, and therefore, in principle, the phase estimation algorithm [30] allows us to estimate the value of $2\theta$. Furthermore, we do not need to prepare an eigenstate of $G$, as the phase estimation will give either $2\theta$ or $-2\theta$ randomly. Estimation of either of the values suffices. We will show shortly that the trivial closeness of a given curve $r$ on the triangulated mesh is encoded in the corresponding phase, and therefore could be revealed by looking at the outcome of the quantum phase estimation algorithm.

## 4.2 Algorithm for Detecting Homology Class of a Closed Curve $r$

We have described our quantum preliminaries in Sec. 3. In our algorithm, all qubits will be divided into five different quantum registers: (1) a single qubit anchor register (denoted by subscript $a$), which plays the same role as the first qubit of Eqn. (7); (2) an orientation flag (denoted by subscript $o$), which is used to indicate the orientation of the edge: $|0\rangle_o|e\rangle_e = |\vec{e}\rangle$ and $|1\rangle_o|e\rangle_e = |-\vec{e}\rangle$; (3) the edge data register (denoted by subscript $e$), such as $|e\rangle_e$ which we have just illustrated in (2); (4) the status register (denoted by subscript $s$), which is used to store the status from querying the oracle about a half-edge; and (5) the extra qubit (labelled by subscript $t$) to implementation rotation corresponding to the status register. We remind that that for each cohomology basis there is an efficient unitary $U_\alpha$ to create the cohomlogy basis state $|\Omega_\alpha\rangle = U_\alpha|0^{\otimes \log_2(E)}\rangle_e = \sum_j \omega_j^{(\alpha)}|e_j\rangle_e/\sqrt{c_\alpha}$ in the edge ($e$) register without the orientation label (see Claim. 1 and Appendix C). Moreover, it is also easy to create a superposition of all edges, $|\mathcal{E}\rangle \equiv \frac{1}{\sqrt{E}}\sum_{i=1}^{E}|e_i\rangle_e = U_E|0\ldots 0\rangle_e = H^{\otimes \log_2(E)}|0\ldots 0\rangle_e$, and we have assumed that the total number of edges (without counting the orientation) is a power of two. (If this is not the case, one can always add 'fictitious' edges to pad the total number to be a power of 2.) The goal is to construct a unitary process $U$ that takes $|0\ldots 0\rangle$ to $(|+\rangle|x\rangle + |-\rangle|y\rangle)/\sqrt{2}$.

We are now ready to describe the procedure of our algorithm:

1. [Initialization] First, we construct the following state:

$$|\varphi\rangle = \frac{1}{\sqrt{2}}\big(|0\rangle_a|-\rangle_o|\Omega_\alpha\rangle_e + |1\rangle_a|+\rangle_o|\mathcal{E}\rangle_e\big), \tag{11}$$

which can be created from $|0\rangle_a|0\rangle_o|0\dots0\rangle_e$ by first applying $H \otimes H$ to $|0\rangle_a|0\rangle_o \to (|0\rangle_a + |1\rangle_a)|+\rangle_o/\sqrt{2}$, followed by a controlled operation $|0\rangle_a\langle0| \otimes Z_o \otimes U_\alpha + |1\rangle_a\langle1| \otimes I_o \otimes U_E$.

We will keep the first qubit $(a)$ in the $|0/1\rangle$ basis until the end, where we will turn it into $|+/-\rangle$. Equivalently, $|\varphi\rangle$ can be rewritten as

$$|\varphi\rangle = \frac{1}{\sqrt{2}}\Big(|0\rangle\frac{1}{\sqrt{2}}\sum_i \omega_i(|\vec{e}_i\rangle - |-\vec{e}_i\rangle) \tag{12}$$

$$+ |1\rangle\frac{1}{\sqrt{2E}}\sum_{i=1}^{E}(|\vec{e}_i\rangle + |-\vec{e}_i\rangle)\Big).$$

2. [Applying oracle]. Next we append the status ancillas $|00..0\rangle_s$ and apply the black-box oracle $\mathcal{O}_r$ to the $e$ register (the $\log_2(E)$-qubit system) plus status register, we obtain:

$$|\varphi_1\rangle \equiv \mathcal{O}_r|\varphi\rangle\,|00..0\rangle_s = \frac{1}{\sqrt{2}}\Big[|0\rangle_a\frac{1}{\sqrt{2c_\alpha}}\sum_i \omega_i\big(|\vec{e}_i\rangle|\mathrm{st}(\vec{e}_i)\rangle_s$$

$$-|-\vec{e}_i\rangle|\mathrm{st}(-\vec{e}_i)\rangle_s\big) + |1\rangle_a\frac{1}{\sqrt{2E}}\sum_i \big(|\vec{e}_i\rangle|\mathrm{st}(\vec{e}_i)\rangle_s$$

$$+|-\vec{e}_i\rangle|\mathrm{st}(-\vec{e}_i)\rangle_s\big)\Big], \tag{13}$$

where $\mathrm{st}(\pm\vec{e}_i)$ is used to denote the status value after the query.

3. [Conditional rotation]. Append an ancilla $|0\rangle_t$ (noting the subscript $t$) to $|\varphi_1\rangle$ and apply a rotation on qubit $t$, conditioned on the anchor qubit $a$ being $|0\rangle_a$ and the degree of rotation on the status qubit $|\mathrm{st}(\pm\vec{e}_i)\rangle_s$, so that: $|0\rangle_t \to |\mathrm{Rot}(\vec{e})\rangle_t \equiv \frac{\mathrm{st}}{K}|0\rangle_t + \sqrt{1 - \mathrm{st}^2/K^2}|1\rangle_t$.

This operation only affects the part entangled with $|0\rangle_a$, which becomes

$$\frac{1}{\sqrt{2c_\alpha}}\sum_i \omega_i\Big(|\vec{e}_i\rangle|\mathrm{st}(\vec{e}_i)\rangle_s\,|\mathrm{Rot}(\vec{e}_i)\rangle_t - \tag{14}$$

$$|-\vec{e}_i\rangle|\mathrm{st}(-\vec{e}_i)\rangle_s\,|\mathrm{Rot}(-\vec{e}_i)\rangle_t\Big). \tag{15}$$

The other part entangled with $|1\rangle_a$ remains unaffected.

4. [Oracle unquery] After the conditional rotation, we uncompute or unquery the status register by applying $\mathcal{O}_r$ one more time, assuming that its operation is addition bitwise. We then arrive at a state of the form $|\varphi_2\rangle = (|0\rangle_a|x\rangle + |1\rangle_a|y\rangle)/\sqrt{2}$. The first part then becomes

$$|x\rangle \equiv \frac{1}{\sqrt{2c_\alpha}}\sum_i \omega_i\Big(|\vec{e}_i\rangle\,|\mathrm{Rot}(\vec{e}_i)\rangle_t$$

$$-|-\vec{e}_i\rangle\,|\mathrm{Rot}(-\vec{e}_i)\rangle_t\Big) \otimes |0\dots0\rangle_s, \tag{16}$$

where we have factorized out the status register at the end. The second part becomes

$$|y\rangle \equiv \frac{1}{\sqrt{2E}} \sum_i (|\vec{e}_i\rangle + |-\vec{e}_i\rangle) \otimes |0\rangle_t \otimes |0\ldots0\rangle_s. \tag{17}$$

Here we can evaluate the inner product of $|x\rangle$ and $|y\rangle$, and we obtain

$$\langle x|y\rangle = \frac{1}{\sqrt{4c_\alpha E}} \sum_i \omega_i \big(\langle \mathrm{Rot}(\vec{e}_i)|0\rangle_t - \langle \mathrm{Rot}(-\vec{e}_i)|0\rangle_t\big)$$

$$= \frac{1}{\sqrt{4c_\alpha E}\,K} \sum_i \omega_i \big[st(\vec{e}_i) - st(-\vec{e}_i)\big]$$

$$= \frac{1}{\sqrt{4c_\alpha E}\,K} \Omega_\alpha(r). \tag{18}$$

Note that $st(\pm\vec{e}_i)$ is counting the number of times a half-edge $\pm\vec{e}_i$ appears on the curve $r$. $K = 2$ reduces to the case where the coefficients in half-edges are in $\mathbb{Z}_2$.

5. [Hadamard test state] Apply the Hadamard gate to the anchor qubit $a$, and we have the final state,

$$|\varphi_3\rangle = \frac{1}{\sqrt{2}} \big[ |+\rangle_a |x\rangle + |-\rangle_a |y\rangle \big],$$

which will be used for the Hadamard test. We denote the whole procedure from step 1 to step 5 as a unitary gate $U$, i.e., $|\varphi_3\rangle = U|0\rangle_a|0\rangle_o|0\ldots0\rangle_e|0\ldots0\rangle_s|0\rangle_t$. What we have achieved here is to translate our problem directly into the Hadamard test formalism described earlier, including the construction of the operator $G = US_0U^\dagger(Z \otimes I^{\otimes N})$, where $N = 2 + n_s + \log_2(E)$ with $n_s$ being the number of qubits in the status register.

If the curve $r$ is closed trivially, which means that the sum $\Omega_\alpha(r)$ along the curve vanishes identically (see Eqn. 1), i.e., $\cos(2 \cdot \theta') = 0$, implying that $2 \cdot \theta' = \pi/2 \Rightarrow \theta' = \pi/4$. If we write $\theta' = 2\pi w' \Rightarrow w' = 1/8$, which can be represented by finite bits (more precisely, 3 bits as $1/8 = 0.001$ in binary fraction). Therefore, ideally, it means that the phase estimation algorithm can output such a value with certainty, which again, can be used to verify whether or not $\Omega_\alpha(r) = 0$. Moreover, in the case that the curve $r$ is not homologically trivial, the value $\Omega_\alpha(r)$ can still be obtained from quantum phase estimation, given sufficient ancillas to encode the phase.

6. [Phase estimation] We run the phase estimation algorithm for the operator $G$. As remarked earlier, it does not require us to prepare an eigenstate of $G$ and the phase estimation procedure will allow us to estimate $\pm 2\theta$, which is sufficient for extracting $\Omega_\alpha(r)$. We then repeat the whole procedure for different cohomology basis state $|\Omega_{\alpha'}\rangle$. A curve that is homologically trivial will have all $\Omega_\alpha(r)|_1^{2g} = 0$. Furthermore, complete information for all $\Omega_\alpha(r)|_1^{2g}$ determines the homology class of the curve $r$. However, whether we can determine nonzero values of $\Omega_\alpha(r)|_1^{2g}$ with sufficient accuracy remains to be checked.

**Analysis of accuracy**. In the homologously trivial case, the phase can be represented exactly with finite bits and our algorithm can return an exact result with certainty. However, we need to distinguish this case from the nontrivial cases and there is indeed a finite gap in the phase between the two cases as we show below. Therefore, $\mathcal{O}(1)$-time running of

quantum phase estimation can already determine whether or not the curve is homologous to zero. More specifically, the analysis of [29] shows that generally, the phase register returns two closest values to our true phase value with high probability ($> 4/\pi^2$). In particular, the success probability of measuring best approximated phase value could be amplified to arbitrarily closed to 1 using additional qubits in phase registers [32]. Therefore, in principle, $\mathcal{O}(2g)$ (since we need to repeat the procedure for all cohomology basis $\{\Omega_i\}$) repetitions are enough to determine if the curve $r$, which is specified by $U_r$, is closed.

Let us further analyze the precision, as in the case of homologously non-trivial curve, the value of angle $\theta'$ might be very closed to $\pi/4$, which means that the outcome of phase estimation circuit with low number of precision bits might not suffice to determine the phase and hence the value $\Omega_\alpha(r)$. If the curve $r$ is homologous to zero, we definitely have $\langle x|y\rangle = 0$. Since the value of $w_i$ is either -1, 0, or 1 (taking the orientation into account), if the curve is not homologously zero, then the minimum absolute value of such overlap is $|\langle x|y\rangle| = 1/(2 \cdot \sqrt{c_\alpha \cdot E} \cdot K)$. More specifically, let:

$$|\cos(2 \cdot \theta_0)| = 0,$$
$$|\cos(2 \cdot \theta_m)| = 1/(2 \cdot \sqrt{c_\alpha \cdot E} \cdot K),$$

where $\theta_0$ refers to the case if the curve is homologically trivial, and $\theta_m$ refers to the smallest angle in the non-trivial case. We apply the following inequality

$$|x - y| \geq |\cos(x) - \cos(y)|, \tag{19}$$

and obtain that $|2(\theta_0 - \theta_m)| \geq 1/(2 \cdot \sqrt{c_\alpha \cdot E} \cdot K)$. This means that there is a gap $\Delta$ between the trivial phase value and non-trivial phase value. To distinguish between those phases, we require our phase estimation algorithm to have the error $\delta \leq \Delta$ (the smaller the better). Therefore, the number of qubits $p$ in the phase register required to have the desired accuracy is $\Omega(\log(1/\delta)) = \Omega(\log(\sqrt{c_\alpha \cdot E} \cdot K))$. We can simply choose $p = \lfloor \log(\sqrt{c_\alpha \cdot E} \cdot K) \rfloor + k$, where $k$ is some integer. We recall that $c_\alpha$ is the number of edges with non-zero values for a given cohomology basis element $\Omega_\alpha$. This number is usually much smaller than $E$, which is the total number of edges on the mesh. We also have the condition that $K$ is bounded above. Assume that the mesh has $M$ vertices, then $E$ is $\mathcal{O}(M)$, which means that $\log(\sqrt{c_\alpha \cdot E} \cdot K)$ is $\mathcal{O}(\log(M))$. Therefore, in our algorithm, as long as we can have enough qubits $\sim \mathcal{O}(\log(M))$ in the phase estimation algorithm, then the accuracy $\Delta$ is always guaranteed.

We summarize our main result with the following theorem.

**Theorem 1 (Homology detection)** *Given a closed triangular mesh $M$ of genus $g$, cohomology basis $\langle \Omega_1, \Omega_2, ..., \Omega_{2g} \rangle$, and quantum oracle $U_r$ that specifies a given closed curve $r$. There exists a quantum algorithm that determines the homology class of $r$ in $\mathcal{O}(2g)$ time.*

In comparison, the naive classical running time is $\mathcal{O}(2g \cdot L)$ (which can be improved to $\mathcal{O}(\log(L))$ by parallelization) where $L$ is the 'size' of the curve $r$, i.e., the number of edges on it, as one needs to sum up all the values $\omega_i$ on the all edges $\vec{e}$ of $r$ for all cohomology basis elements $\Omega_{\alpha=1...2g}$. Our quantum algorithm substantially improve the time complexity from linear (in $L$) to constant. We note that our algorithm works even if the given curve has a self-crossing, which can be regarded as a sum of 1-cochains that are loops.

If two closed curves corresponding to two chains $\sigma_1$ and $\sigma_2$ are homologous, i.e., in the same equivalence class, then they differ by an exact 2-chain. This means for any 1-cochain

$\omega$, we have $\omega(\sigma_1) = \omega(\sigma_2)$. If we decompose the 1-cochain $\omega$ into the basis $\{\Omega_i\}_{i=1}^{2g}$ then for each basis element we need to have $\Omega_i(\sigma_1) = \Omega_i(\sigma_2)$. We can compute $\Omega_i(\sigma_1)$ and $\Omega_i(\sigma_2)$ directly and separately using our quantum algorithm (given the respective oracles) to compute $\Omega_i(\sigma_{1/2})$ and check if they equate each other for all $i = 1, \ldots, 2g$.

## 4.3 Potential Applications

### 4.3.1 Homotopy Detection

Given the ability to efficiently determine the homology class of a closed curve, it is natural to seek applications of the quantum algorithm. Here we point out an instance that also arises in the computational conformal geometry and topology context [5,7], i.e. the so-called homotopy detection. The statement of the problem is the following.

**Problem 1 (Homotopy Detection)** *Given a closed triangular mesh M, two loops $\gamma_1$ and $\gamma_2$ through a base point p. Verify whether or not $\gamma_1$ is homotopic to $\gamma_2$, $\gamma_1 \sim \gamma_2$.*

Such a problem also has a linear time classical solution [33]. While homology groups are commutative, homotopy groups are usually non-commutative; therefore, homotopy can generally be harder to deal with than homology, such as computing its groups. There are also hard problems related to homotopy. For example, the shortest word problem [34] for a given homotopy class is NP-hard [35]. While we do not know whether we could completely solve the homotopy detection problem with a constant time algorithm, we observe an essential property that, *two loops are homotopic to each other implying that they are homologous* (the reverse may not be true). In particular, in the homotopy detection problem, if $\gamma_1 \sim \gamma_2$, then $\gamma = \gamma_1 \cdot \gamma_2^{-1}$ is homotopic to $e$, i.e, the loop is trivial (constant loop). Thus, if $\gamma \sim e$, then $\gamma$ is necessarily homologous to 0 as well. As we can only check whether the curve is homologous to 0, we can apply our algorithm to check the converse, i.e., we can ascertain the case when two curves (on a closed surface) are *not* homotopic to each other by verifying that the loop $\gamma$ is not homologous to zero.

In Ref. [6], an alternative classical solution to the homotopy detection problem was described. The key idea is that, we first compute a finite portion of the universal covering space $\tilde{M}$ of $M$. We then lift $\gamma_1 \cdot \gamma_2^{-1}$ to $\tilde{M}$, and denote the lifted path as $\gamma$. If $\gamma$ is a loop, then $\gamma_1 \sim \gamma_2$. However, converting the above solution into an efficient quantum algorithm is an open problem.

### 4.3.2 Winding Number Estimation

It is pretty interesting that aside from the homology detection problem, the algorithm that we use in this paper (integration of cohomology basis) can be employed to estimate the numbers of times that a loop winds around the torus.

In Fig. 3, denote the homology and cohomology basis as $(h_1, h_2)$ and $(\Omega_1, \Omega_2)$ (see Fig. 2a). W.O.L.G., let $h_1$ be the red curve, and $h_2$ be the blue curve. The integration of cohomology basis along the winding loop is:

$$\int_r \Omega_1 = 1, \quad \int_r \Omega_2 = m, \tag{20}$$

where $m$ is the number of times that the given loop $r$ winds around the torus. Given an oracle, the algorithm developed in Sec. 4.2 can estimates the above integral (more precisely, a

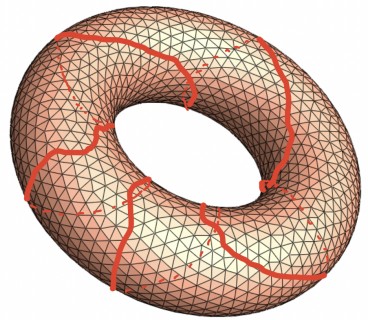

Figure 3: A loop r winds around the torus 5 times.

discrete summation). Therefore, our algorithm can estimate the winding numbers of the given loop $r$. In this case, as shown in Fig. 3, no half-edge appears twice, therefore, the simpler $\mathbb{Z}_2$-version of the oracle (i.e., $K = 2$) is sufficient.

## 5 Discussion and Conclusion

Topology and geometry are very rich and deep area of mathematics and their applications to science and engineering have been increasingly broadened due to the connection from the profound mathematical foundation to classical computational geometry algorithms. This, in turn, yields powerful tools to solve many practical problems. Quantum computer is also undergoing a second wave of fast development, and concurrently, the potential power and applicability of quantum computers have always been an important question to address. Our work has added to the few existing works and extended the application of quantum computation to computational topology, in particular, the problem of homology detection. Our quantum method relies on important observation: cohomology assigns real values to half-edges of a mesh, which could be stored efficiently using a logarithmic number of qubits. The summation along a loop (which is specified by an oracle) is done using the orthogonal relation of basis states $|e_i\rangle$. Key tools include the Hadamard test, amplitude amplification, and quantum phase estimation.

Even though the particular problem we have considered is not a hard one from classical complexity's perspective (e.g., with a running time $\mathcal{O}(\log(L))$ in the number $L$ of edges on the curve), our quantum algorithm achieves a constant time complexity, yielding a substantial speedup. We note that there are other problems where classical solutions are efficient, but the quantum algorithms are even more efficient, such as the 2D hidden linear function problem, where the classical solution requires at least a logarithmic depth whereas the quantum solution requires a shallow depth [25]. This problem was further generalized to yield an exponential separation between the classical fan-out circuits and shallow-depth quantum circuits [36].

We have also pointed out the potential application of our quantum algorithm in the homotopy detection problem. As described above, the homotopic relation implies homology relation, therefore, we could check the homology condition as a mean to rule out the non-homotopic relation. We have also suggested an open problem regarding how to construct a quantum algorithm that elevates the (classical) universal-covering-space approach provided in [6].

We now discuss an important open point of our work, which appears in most oracle-

based quantum algorithms, including Grover's search, namely, the construction of the oracle itself. We have assumed that the oracle $\mathcal{O}_r$ could efficiently query the half-edge, but left aside the detail of such an oracle. How to implement the oracle explicitly (and efficiently) is an open question. We remark that a mesh is mathematically a graph, therefore, and we could in principle use the graph Laplacian to encode the necessary information of the mesh (i.e., connectivity), for example, in a QRAM [37], then the connectivity (plus proper orientation) could be accessed coherently. If an explicit construction for the oracle $\mathcal{O}_r$ were known, then we could deal with an arbitrary curve on the graph, which in turn might provide more flexibility for applications.

Homology and homotopy detection are critical for a broad range of applications. In medical imaging, for example, virtual colon cancer detection, the colon surface is reconstructed from CT-images. Due to the segmentation error, there are many fake handles and tunnels on the reconstructed surfaces. It is crucial to detect these spurious topological noises and remove them by topological surgery using homological method [38, 39]. In wireless sensor network field, homology is applied for coverage and hole detection in sensor networks [40] and location tracking [41]. It is interesting to see potential applications of quantum algorithms on these practical problems. As a final remark, our work mainly deals with the first homology group $H_1$ and it is natural to ask how to extend our algorithm to deal with arbitrary homology group $H_k$ $(k > 1)$, which is left for future exploration.

# Acknowledgements

The research reported in this work was partially supported by the National Science Foundation under Grants No. PHY 1915165 (T.-C.W.), No. 2115095 (X.D.G.), and No. 1762287 (X.D.G.), as well as NIH R21EB029733 (X.D.G.).

# A  More on homology/cohomology basis

Here we elaborate further on the concept of cohomology basis and its computation. For more details, we refer the readers to [6, 7]. For illustration, we denote the red and blue curves in Fig. 2(a) as $\gamma_1$ and $\gamma_2$, respectively. We remark that the cohomology group contains the equivalence classes of closed cochain, which could be constructed from homology basis as follows. We slice the whole mesh along, for example, $\gamma_1$ and obtain an open mesh with two boundaries $\gamma_1^+$ and $\gamma_1^-$. We then initialize 0-form $f_i$ that satisfies: $f_i(v_j) = 1$ for $v_j \in \gamma_1^+$ and 0 otherwise. With this initialization, $df_i$ is then closed but non-exact on the original mesh, hence forms the first cohomology basis. The same process yields the remaining cohomology basis. From such construction we easily notice that with a given cohomology basis $df_i$, $df_i(e_j)$ is then 0 everywhere except those 'half-edges' $e_j$ that connect a point $v_i \in \gamma_i$ and a neighboring $v_k \notin \gamma_i$. We note that the two half-edges of an edge correspond to its two 'sides' with opposite orientations; but we do not need to invoke half-edges in the main text of this paper.

## A.1  Encoding of Cohomology Basis in a Quantum State

We remark that cohomology assigns each edge (with specified orientation) a real number, therefore, the set of those numbers on given mesh $M$ can be stored in a quantum state (more

precisely, a vector). Suppose we are given some 1-cochain $w$, and can define a corresponding quantum 1-cochain:

$$|w\rangle = \sum_{i=1}^{E} w_i |e_i\rangle / \sqrt{\sum_i w_i^2}, \tag{21}$$

where $w_i = w(\vec{e_i})$ (note the orientation) and $E$ is the total number of edges. In general, it is nontrivial to create such a quantum state. However, for the cohomology basis states $|\Omega_\alpha\rangle$s, many of the coefficients $w_i$'s are zero and those nonzero ones can be chosen to be unity and there is an efficient circuit to construct these states, see Appendix C.

## B  Elaboration of The Main Method

We remind that the main tool used for detecting closed curves was based on the integral (or summation in our discrete case) of each cohomology basis $\{\Omega_\alpha\}$ along such a curve $r$, and we require all of them to be 0, if $r$ is homologically trivial. Here we explain further why we require such a condition.

A $k$-chain $\sigma_k$ is said to be *exact* if it is the image of some (k+1)-chain under the boundary map $\partial_{k+1}$

$$\partial_{k+1}\sigma_{k+1} = \sigma_k \tag{22}$$

As a dual relation between homology and cohomology, for each boundary operator $\partial_k$, there is a co-boundary operator $d^k$ that maps a $k$-cochain to a $(k+1)$-cochain. Let $\omega'$ be a $(k-1)$-cochain, $\sigma_k$ be a $k$-chain, we have the following property:

$$d^{k-1}\omega'(\sigma_k) = \omega' \cdot (\partial_k \sigma_k) \tag{23}$$

Now we apply (closed) k-cochain $\omega^k$ on both sides of Eqn. 22:

$$\omega^k(\partial_{k+1}\sigma_{k+1}) = \omega^k \sigma_k \tag{24}$$

Employing the property in B2, we have:

$$\omega^k(\partial_{k+1}\sigma_{k+1}) = d^k \omega^k \sigma_{k+1} \tag{25}$$

Since $\omega^k$ is closed, $d^k \omega^k = 0$, therefore, it means that $\omega^k \sigma_k = 0$. Set $\omega^k \equiv \omega$ for brevity.

Now we use an important property of the cochain space: it is a linear vector space, and therefore, a given cochain $\omega$ can be decomposed into a set of bases. Denote the set of basis as $\{\omega_i\}_{i=1}^k$. We then have: $\omega = \sum_i a_i \omega_i$, where each $a_i$ is generally a complex component. We now obtain the following:

$$\sum_i a_i \omega_i(\sigma_k) = 0. \tag{26}$$

We remark that as the basis $\{\omega_i\}_{i=1}^k$ is linearly independent, therefore, in order for the above sum to be 0, we need every term to be 0, and since $\{a_i\}$'s generally are not zero, this implies that, for each $i$,

$$\omega_i \cdot (\sigma_k) = 0. \tag{27}$$

In our problem, we are interested in 1-chains, i.e. $k = 1$. The summation along the curve is simply the summation along all the (half-)edges on the boundary.

# C Details on Cohomology Basis State Preparation

Here we elaborate further the claim 1, which is probably the most crucial part of the algorithm. Uniform superposition of data is useful in quantum computation, or more specifically, in quantum machine learning, where s amplitude encoding is one of the most popular approaches to load classical data into a quantum state [18, 42, 43].

We remind that the geometrical object is represented as a triangular mesh in the discrete setting. Therefore, there is naturally a hardware structure equipped with the object. In our case specifically, it is the (ordered) set of vertices, edges and faces. In general, cohomology theory deals with a linear vector space. The key idea of our quantum algorithm is to map cochains to quantum states, so as to leverage the quantum resources. A subtle point in the construction of cohomology basis state as in Claim 1 is the order of those edges that we impose on via classical processing (we have remarked that the mesh generation process is done classically). For each cohomology basis $\vec{\Omega}_\alpha$ (or we can write $|\Omega_\alpha\rangle$ in quantum setting), there are $c_\alpha$ non-zero entries, and these entries have consecutive order. As we mentioned, the cohomology basis vector has the following form:

$$\vec{\Omega}_\alpha = [0, ..., \pm 1, \pm 1, ...0]^T$$

with $c_\alpha$ non-zero entries. The corresponding quantum state is simply

$$|\Omega_\alpha\rangle = \vec{\Omega}_\alpha / \sqrt{c_\alpha}.$$

The reason why we want to order those edges (the order of those non-zero entries) in consecutive order is to achieve a low-cost way to prepare such a state, which we describe now.

**An efficient algorithm for uniform superposition**. First, we argue that we can always label edges in the cohomology basis states such that they are consecutive in number. We argue that two different cohomology basis states can have only one overlapping edge and that it can be arranged so that for any one cohomology basis state, there is at most one other cohomology basis state that has an overlapping edge. Therefore, all the edges in involved in all cohomology basis states can be labeled consecutively. If there is an overlap, then the last edge of the cohomology basis state is the first edge of the next one that has an overlap.

Once the labelling is settled, then we need to show that we can create an equal superposition of basis states that are binary representation of consecutive numbers, i.e., $|\psi\rangle = \sum_{j=a_1}^{a_2} |j\rangle / \sqrt{a_2 - a_1 + 1}$ can be created efficiently. We then argue that this reduces to creating $|\psi_0\rangle = \sum_{j=0}^{a_2-a_1} |j\rangle / \sqrt{a_2 - a_1 + 1}$, as there is an efficient adder (of $\log(|a_2 - a_1 + 1|)$ depth); see e.g., Ref. [28]. Namely by adding $|\psi_0\rangle$ to $|a_1\rangle$.

**An example**. Next, we will focus on creating $|\psi_0\rangle$ and describe an efficient way to create some superposition of $|0...0\rangle$ to $|a_n, a_{n-1}, ..., a_0\rangle$. Let us first illustrate this by an example. Suppose we want to make a superposition from 0000 to 1011 (and we will take the last configuration 1011 as a reference, so we can apply gates conditioned on some of the digits).

First, we start with $|0000\rangle$ and, recognizing the most significant digit of '1011' is one, we create a superposition $c_0|0000\rangle + d_0|1000\rangle$ (denoting this operation by UIII), followed by a product of Hadamard gates conditioned on the first quantum bit being 0: $|0\rangle\langle 0| \otimes H_2 \otimes H_1 \otimes H_0 + |1\rangle\langle 1| \otimes I \otimes I \otimes I$, with the operation denoted by 0HHH. Thus, we obtain a state $d_0|1000\rangle + c_0|0 + ++\rangle$.

Then, we move down the bit string in '1011' and it is '0', so we do nothing in this step. (If it were '1' we would split 1000 to $c|1000\rangle + d|1100\rangle$.) We move down the bit string

again to reach '1', and we split (conditioned on the significant qubit being in $|1\rangle$) 1000 to $c_2|1000\rangle + d_2|1010\rangle$ (i.e. an operation labeled as 10UI), followed by a controlled gate 100H (i.e., a Hadamard gate conditioned on the first three qubits being in $|100\rangle$ state), where the first two bits '10' are from the string '1011' and the '0' before H is fixed. And we arrive at $c_0|0{+}{+}{+}\rangle + d_0(c_2|100{+}\rangle + d_2|1010\rangle)$. Finally, we reach the last digit of '1011', which is '1'. If it were zero, we would do nothing. But given this is '1', we perform an operation controlled on the first three qubits being 101 (denoting this operation as 101U) to take $|1010\rangle$ to $c_3|1010\rangle + d_3|1011\rangle$, so the final state is $c_0|0{+}{+}{+}\rangle + d_0(c_2|100{+}\rangle + d_2(c_3|1010\rangle + d_3|1011\rangle))$. We can check the number of computational-basis terms is $2^3 + 2^1 + 2^0 + 1 = 12$ and it contains the all the desired components. If we adjust the coefficients appropriately, we can obtain a uniform superposition from $|0000\rangle$ to $|1011\rangle$. In summary, the operations are: {(UIII, 0HHH), (10UI, 100H), (101U)} acting on the initial $|0000\rangle$ state.

**General case**. So the algorithm for creating a superposition of components from $|0\ldots0\rangle$ to $|a_n, a_{n-1}, \ldots, a_0\rangle$ is as follows. Start with the $|0\ldots0\rangle$ state (if there are more qubits than $n+1$, then pad the remaining qubits to $|0\rangle$ and the following procedure applies to the relevant part of the qubits). One reads the bit string $a_n a_{n-1} \ldots a_0$. Begin with $k = n$. If $a_k = 0$, then one moves on to the next bit. If $a_k = 1$, split $|a_n, a_{n-1}, \ldots, 0_k, \ldots, 0\rangle$ into a superposition of $|a_n, a_{n-1}, \ldots, 1_k, \ldots, 0\rangle$ and $|a_n, a_{n-1}, \ldots, 0_k, \ldots, 0\rangle$ (via operation $a_n, a_{n-1}, \ldots U_k I \ldots I$) followed by the operation $a_n, a_{n-1}, \ldots 0_k H \ldots H$. Iterate this (decrease the index $k$ by one) until we read $a_0$. If $a_0 = 0$, nothing needs to be done. If $a_0 = 1$, then we split the state $|a_n, a_{n-1}, \ldots, a_1, 0\rangle$ via operation $(a_n, a_{n-1}, \ldots, a_1, U)$ into a superposition of $|a_n, a_{n-1}, \ldots, a_1, 0\rangle\rangle$ and $|a_n, a_{n-1}, \ldots, a_1, 1\rangle$. We note the coefficients (that determine the superposition by $U$'s) can be computed beforehand so the final output state is a uniform superposition.

The time complexity is proportional to the number of qubits nontrivially involved in the superposition, the procedure of applying gates follows checking the bits sequentially, i.e., the complexity is roughly $\log_2(|a_2 - a_1 + 1|) = \log_2(n)$. Generalizing this to arbitrary bit strings, we have an efficient algorithm for creating uniform superposition.

**An alternative approach**. We claim that one can always modify the mesh and triangulation so that the number of nontrivial edges in every cohomology basis state is a power of 2. Then we only need to use Hadamard gates and additionally the quantum addition gate.

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
