# Peer review of "Constant-time Quantum Algorithm for Homology Detection in Closed Curves"

_SciPost Physics_

## Round 1 · Referee Report · Anonymous (Referee 1) · 2023-1-3

Strengths

The paper presents a quantum algorithm for deciding whether a given 1-cycle (given in the form of an oracle) of length L on a triangulated Riemann surface of genus g is homologically trivial or nontrivial.

1) The algorithm runs in O(2g) time, independent on the length of the cycle. This is a significant improvement over O(2g L), provided by the best known classical algorithm by Yau et al.

2) The problem is interesting, well motivated by applications, and timely; There has been a large amount of recent activity in quantum algorithms for topological problems and in searching for exponential advantage for large-dimensional holes, including the recent result that the k-dimensional homology problem is QMA1-hard and under certain assumptions contained in QMA, suggesting that homology is closely related to quantum mechanics. This provides an important background for the paper, making the study of quantum algorithms for this problem well motivated.

3) The main technical contribution of the paper seems to be the repurposing and generalization of techniques previously used in quantum neural networks to the problem of homology. To the best of my knowledge this is a new set of techniques, different from those used in the LGZ algorithm and follow-ups for example. The existence of a gap in the phase theta is also an interesting result and crucial for the algorithm to be efficient.

Weaknesses

The paper is generally strong. However, I believe the following questions/comments should be addressed.

1) In the section Homology and cohomology basis, it is mentioned that the values of E and c_\alpha are determined by a certain amount of classical pre-processing. Can the authors elaborate on the complexity of the pre-processing? How does it compare to the running time of the quantum part of the algorithm? Is this pre-processing also assumed in the algorithms by Yau et al.?

2) Another possible (and seemingly much simpler) quantum algorithm for the problem described would be to construct the combinatorial Laplacian, \Delta_1, of the simplicial complex and apply phase estimation to \Delta_1, with the 1-cycle r as an input. The 1-cycle r is closed if and only if it’s eigenvalue under \Delta_1 is zero. Whether this phase estimation procedure is efficient or not depends on the gap of \Delta_1. Can the authors comment on whether they expect this gap and the gap they describe in the angle theta are expected to be related?

3) It would be welcome if the authors could give an estimate of the resources needed to run their algorithm for real-world data sets. Is the algorithm long-term or NISQ?

Report

I believe the problem considered is interesting, timely, and with potential useful applications. The introduction of new algorithmic techniques from quantum neural networks to the study of topology is welcome and could potentially be of more broad applicability.

However, before recommending the paper for publication I would like to see the points above addressed and the many typos/misprints corrected.

Requested changes

A few typos and formatting issues (these are not exhaustive, as they are many):

1) Below first unnumbered equation, it is claimed that exact chains are closed but not explained why.

2) Punctuation after unnumbered equation above equation 2 should be changed to comma.

3) Page 4: cohomology basi -> cohomology basis AND of represent -> to represent

4) Punctuation in Eq. (5)

5) Page 7: “they differ by an exact 2-chain” -> : “they differ by an exact 1-chain, i.e., the boundary of a 2-chain”

  • validity: high
  • significance: high
  • originality: high
  • clarity: good
  • formatting: good
  • grammar: reasonable

Author:  Nhat A. Nghiem Vu  on 2023-01-20  [id 3252]

(in reply to Report 1 on 2023-01-03)

Strengths: The paper presents a quantum algorithm for deciding whether a given 1-cycle (given in the form of an oracle) of length L on a triangulated Riemann surface of genus g is homologically trivial or nontrivial. 1) The algorithm runs in O(2g) time, independent on the length of the cycle. This is a significant improvement over O(2g L), provided by the best known classical algorithm by Yau et al. 2) The problem is interesting, well motivated by applications, and timely; There has been a large amount of recent activity in quantum algorithms for topological problems and in searching for exponential advantage for large-dimensional holes, including the recent result that the k-dimensional homology problem is QMA1-hard and under certain assumptions contained in QMA, suggesting that homology is closely related to quantum mechanics. This provides an important background for the paper, making the study of quantum algorithms for this problem well motivated. 3) The main technical contribution of the paper seems to be the repurposing and generalization of techniques previously used in quantum neural networks to the problem of homology. To the best of my knowledge this is a new set of techniques, different from those used in the LGZ algorithm and follow-ups for example. The existence of a gap in the phase theta is also an interesting result and crucial for the algorithm to be efficient. [Our response]

We thank the reviewer for their positive comments on our work. We would like to remark that the naive calculation of a cohomology basis over a loop takes linear time in the loop size L. However, one can improve it to log(L) by a parallel algorithm. Nevertheless, our constant time complexity is still an significant improvement over log(L) time complexity and we have updated that in our manuscript.

[end response]

Weakness:

1) In the section Homology and cohomology basis, it is mentioned that the values of E and c_\alpha are determined by a certain amount of classical pre-processing. Can the authors elaborate on the complexity of the pre-processing? How does it compare to the running time of the quantum part of the algorithm? Is this pre-processing also assumed in the algorithms by Yau et al.?

[our response]

Both classical and quantum algorithms require mesh pre-processing (which takes linear time in the number of triangles, proportional to the total number of edges |E|); in particular, the homology basis/ cohomology basis construction is needed (also assumed in the algorithm by Gu and Yau). The pre-processing is independent of the curves to be analyzed and only needs to be performed once and it takes much longer than that the running time of the quantum part.

[end response]

2) Another possible (and seemingly much simpler) quantum algorithm for the problem described would be to construct the combinatorial Laplacian, \Delta_1, of the simplicial complex and apply phase estimation to \Delta_1, with the 1-cycle r as an input. The 1-cycle r is closed if and only if it’s eigenvalue under \Delta_1 is zero. Whether this phase estimation procedure is efficient or not depends on the gap of \Delta_1. Can the authors comment on whether they expect this gap and the gap they describe in the angle theta are expected to be related?

[our response]

We thank the reviewer for the interesting suggestion. However, the combinatorial Laplacian acting on all closed curves will give zero. Meanwhile there are two types of closed curves: exact and non-exact, and combinatorial Laplacian cannot distinguish between the two, which is the main problem we deal in our paper. So we do not think it can be used to detect homology. However, it may be used to detect whether a curved is closed or not.

Moreover, simulating the combinatorial Laplacian would take a significant amount of time, because the dimension of combinatorial Laplacian grows at least polynomially with the number of simplexes in the mesh.

Classically, in order to compute the spectrum of the combinatorial Laplace matrix, one needs to compute the Smith norm form of the integer matrix, which is NP-hard in general. The gap in Laplace-Beltrami operator is the Cheeger’s constant, which is determined by the topology and the Riemannian metric of the manifold. The geometric meaning of this gap is as follows: Let M be a n-dimensional closed Riemannian manifold, Let S(E) denote the volume of an (n-1)-dimensional submanifold E, the Cheeger isopermetric constant of M is the inf of S(E)/min(V(A),V(B)), where the infinimum is taken over all the (n-1)-dimensional submanifold E of M which divides M into two disjoint submanifolds A and B. In the 2D surface case, E is the shortest loop which divides the surface into two halves with equal area. Therefore, we do not expect the gap in the combinatorial Laplacian is related to our gap (angle theta). Our gap is comes from the accuracy between the trivial closed curves and nontrivial-closed curves.

[end response]

3) It would be welcome if the authors could give an estimate of the resources needed to run their algorithm for real-world data sets. Is the algorithm long-term or NISQ?

[our response]

This is a longer-term quantum algorithm. A NISQ version will likely be in the form of a variational quantum eigensolver. Resource estimation: as the algorithm runs in constant time, the key resource requirement is the number of qubits (i.e. memory) needed and the execution of phase estimation and the Hadamard test (which requires long coherence). Specifically, the number of qubit depends logarihmically on the number edges. If we wish QPE’s accuracy to be epsilon, then we need the number of qubit to scale as log (1/epsilon), where epsilon is the error. QPE puts a stringent requirement on the gate fidelity, as we need long-range controlled gates for the execution of QPE

[end response]

Anonymous on 2023-04-03  [id 3535]

(in reply to Nhat A. Nghiem Vu on 2023-01-20 [id 3252])
Category:
remark
objection
correction

The authors have said, in response to my question that:

“ We thank the reviewer for the interesting suggestion. However, the combinatorial Laplacian acting on all closed curves will give zero. Meanwhile there are two types of closed curves: exact and non-exact, and combinatorial Laplacian cannot distinguish between the two, which is the main problem we deal in our paper. So we do not think it can be used to detect homology. However, it may be used to detect whether a curved is closed or not.”

This is an absolutely incorrect statement. I apologize for being blunt, but it worries me such a basic misunderstanding may signal that the authors don’t have a good grasp of basic aspects of homology, which is the main subject of their paper.

Indeed, as the authors may read in any introductory text to homology, the combinatorial laplacian vanishes on a curve only if the curve is closed AND if that cycle is not exact. Thus, the Laplacian can be used to detect holes and indeed this is the main property exploited in the LGZ algorithm, which the authors cite.

I thus invite the authors to improve their understanding of this rather basic point and to provide a new answer to the question originally posed in my first review.

Anonymous on 2023-04-07  [id 3563]

(in reply to Anonymous Comment on 2023-04-03 [id 3535])

We thank the reviewer for pointing out the flaw in our previous comment and we are terribly sorry for making such a mistake. Indeed, the eigenvectors of the combinatorial Hodge Laplace operator corresponding to the zero eigenvalue are closed and non-exact chains, namely they span the H_k(M,Z) for any k. This is a classical algebraic method for computing the homology group, which involves finding the Smith normal form of integer matrices.

While the combinatorial Laplacian method seems simpler at first sight, a critical issue that may arise is the gap between eigenvalues of the combinatorial Laplacian, which may be very small. In our previous answer, we already provided some geometrical meaning of such a gap and described the difficulty to estimate it, and that statement still holds. Here we would like to emphasize further that, in reference [1], the authors also discussed the same issue with the gap of the combinatorial Laplacian, and mentioned that an exponentially small gap is possible in general, which would induce a very substantial running time if we wish to use the combinatorial Laplacian approach.

In summary, despite our embarrassing comment and its apparent mistake, we would like to emphasize that our answer to the question raised by the reviewer on the relation of the two gaps still holds, namely, the gap from combinatorial Laplacian and the gap in the phase estimation (the latter is finite) discussed in our work are not related.

Reference:
[1] Schmidhuber, Alexander, and Seth Lloyd. "Complexity-Theoretic Limitations on Quantum Algorithms for Topological Data Analysis." arXiv preprint arXiv:2209.14286 (2022)

---

## Round 1 · Referee Report · Anonymous (Referee 2) · 2023-2-27

Strengths

The results in this paper are interesting and well-motivated. The main problem addressed in this paper is to determine whether a given closed curve or loop on a closed surface, represented as a triangular mesh, is homologically trivial. The ability to detect such curves is important to homotopy detection and various topological problems. The paper draw an interesting connection between topology and quantum algorithms. While there exists an efficient classical algorithm for this problem, the development of even faster quantum algorithms is highly desirable and this paper presents one such scenario.

Weaknesses

  1. Since the authors consider surfaces as represented by a triangular mesh, it is then related (but not equivalent) to a triangulation of the surface, as a cellular embedding of a graph. It would be worthwhile to explicitly explain how their results relate to other results. In topological graph theory, it is known classically that determining if a closed curve is homologically trivial (or contractible) has a polynomial time solution, however the problem of determining if of a curve is splitting (a cycle is splitting if it is simple, surface separating and not homologically trivial) is NP-hard.

Report

This paper meets SciPost Physics's acceptance criteria. The algorithm presented here is novel and opens a new link between topology and geometry and quantum algorithms, which has much potential for follow-up work and for building the connections between these very different research areas.

Requested changes

  1. Citation [7] is given as arXiv paper, but the paper is published and the peer-reviewed version should be cited/used. (appears in Communications in Information and Systems, Volume 2 (2002), Number 2).

  • validity: high
  • significance: high
  • originality: high
  • clarity: good
  • formatting: good
  • grammar: perfect

Author:  Nhat A. Nghiem Vu  on 2023-03-24  [id 3506]

(in reply to Report 2 on 2023-02-27)

Strengths The results in this paper are interesting and well-motivated. The main problem addressed in this paper is to determine whether a given closed curve or loop on a closed surface, represented as a triangular mesh, is homologically trivial. The ability to detect such curves is important to homotopy detection and various topological problems. The paper draws an interesting connection between topology and quantum algorithms. While there exists an efficient classical algorithm for this problem, the development of even faster quantum algorithms is highly desirable and this paper presents one such scenario.

[Our Response] We thank the reviewer for positive comments on our work. [end response]

Weaknesses 1. Since the authors consider surfaces as represented by a triangular mesh, it is then related (but not equivalent) to a triangulation of the surface, as a cellular embedding of a graph. It would be worthwhile to explicitly explain how their results relate to other results. In topological graph theory, it is known classically that determining if a closed curve is homologically trivial (or contractible) has a polynomial time solution, however the problem of determining if of a curve is splitting (a cycle is splitting if it is simple, surface separating and not homologically trivial) is NP-hard.

[Our response] We thank the reviewer for interesting suggestions. While in general, we don’t expect quantum computers to solve NP-hard problems, however, as the reviewer mentioned, a property of a splitting curve is that it is not homologically trivial. Such property can be verified (in constant time) using our quantum algorithm. In fact, we have mentioned the application of our method to the homotopy detection problem, where the curve, if verified not to be homologically trivial, then it cannot be homotopic to zero. It is quite interesting that at the same time, we can also apply such properties to curve splitting problems, even though our algorithm does not solve it in general. [end response]

Report This paper meets SciPost Physics's acceptance criteria. The algorithm presented here is novel and opens a new link between topology and geometry and quantum algorithms, which has much potential for follow-up work and for building the connections between these very different research areas.

[our response] We thank the reviewer for assessing that our paper meets the acceptance criteria and recognizing that it opens a new link between two fields.. [end response]

Requested changes 1. Citation [7] is given as arXiv paper, but the paper is published and the peer-reviewed version should be cited/used. (appears in Communications in Information and Systems, Volume 2 (2002), Number 2). [Our response] We have updated the change. The new version of citation [7] is now used. [end response]

Anonymous on 2023-03-24  [id 3507]

(in reply to Nhat A. Nghiem Vu on 2023-03-24 [id 3506])

Regarding the Citation [7], we found out that the peer-reviewed version of that paper is not published as a standard article, and therefore we cannot cite the version that the referee mentioned in Communications in Information and Systems, Volume 2 (2002), Number 2. In fact, the result of citation [7] is contained in a standard textbook in that area
"Computational Conformal Geometry", Xianfeng David Gu and Shing-Tung Yau . Therefore, we've used the book as an alternative selection.

---

## Editorial Decision

resubmitted